# The Corpus Adiposum Infrapatellare (Hoffa’s Fat Pad)—the Role of the Infrapatellar Fat Pad in Osteoarthritis Pathogenesis

**DOI:** 10.3390/biomedicines10051071

**Published:** 2022-05-05

**Authors:** Sebastian Braun, Frank Zaucke, Marco Brenneis, Anna E. Rapp, Patrizia Pollinger, Rebecca Sohn, Zsuzsa Jenei-Lanzl, Andrea Meurer

**Affiliations:** 1Department of Orthopedics (Friedrichsheim), University Hospital Frankfurt, Goethe University, 60528 Frankfurt am Main, Germany; marco.brenneis@kgu.de (M.B.); andrea.meurer@kgu.de (A.M.); 2Dr. Rolf M. Schwiete Research Unit for Osteoarthritis, Department of Orthopedics (Friedrichsheim), University Hospital Frankfurt, Goethe University, 60528 Frankfurt am Main, Germany; frank.zaucke@kgu.de (F.Z.); annaelise.rapp@kgu.de (A.E.R.); p.p.pollinger@googlemail.com (P.P.); rebecca.sohn@kgu.de (R.S.); zsuzsa.jenei-lanzl@kgu.de (Z.J.-L.)

**Keywords:** infrapatellar fat pad, knee osteoarthritis, total knee replacement, fat pad resection, inflammation

## Abstract

In recent years, the infrapatellar fat pad (IFP) has gained increasing research interest. The contribution of the IFP to the development and progression of knee osteoarthritis (OA) through extensive interactions with the synovium, articular cartilage, and subchondral bone is being considered. As part of the initiation process of OA, IFP secretes abundant pro-inflammatory mediators among many other factors. Today, the IFP is (partially) resected in most total knee arthroplasties (TKA) allowing better visualization during surgical procedures. Currently, there is no clear guideline providing evidence in favor of or against IFP resection. With increasing numbers of TKAs, there is a focus on preventing adverse postoperative outcomes. Therefore, anatomic features, role in the development of knee OA, and consequences of resecting versus preserving the IFP during TKA are reviewed in the following article.

## 1. Introduction

The infrapatellar fat pad (IFP), which is also referred to as corpus adiposum infrapatellare or Hoffa’s fat pad, named after the famous surgeon to first describe the anatomical structure (Albert Hoffa in 1904) [1], has generated increasing scientific interest in recent years. A causal relationship between the development and progression of knee OA and IFP through extensive interactions with synovium, articular cartilage, and subchondral bone has been presumed and discussed. In the past, IFP has been of interest to orthopedic surgeons primarily as the anatomical site of Hoffa’s syndrome (Hoffaitis)—a post-traumatic condition that constitutes one of the many causes of anterior knee pain [1]. Nowadays, with the continuously growing numbers of TKA, the intraoperative handling of the IFP is at the center of controversies. In this context, IFP-preserving techniques and (partial) IFP resection procedures are being discussed [2]. In the present article, the IFP as an intra-articular structure in the knee joint is introduced with regard to anatomical facts and functional aspects, followed by a thorough discussion of the effects of the respective surgical procedures on postoperative clinical outcomes.

## 2. Methodology

For this narrative review, we analyzed the PubMed bibliographic database using the following keywords: ‘infrapatellar fat pad’, ‘total knee arthroplasty’, ‘molecular mechanism’, ‘osteoarthritis pathogenesis’, ‘resection’ and ‘preservation of infrapatellar fat pad’. In particular, several unrestricted free-text searches were performed combining those terms with AND. No limits of time were adopted, though we reviewed these publications with special emphasis on recent publications. Other reviews and original articles were obtained by consulting the relevant references in these papers.

## 3. Anatomy and Function of IFP

### 3.1. Anatomy of the IFP

The IFP is one of three fat pads in the anterior knee joint (alongside the suprapatellar fat pad and the femoral fat pad). Localized intraarticularly and extrasynovially, the IFP is abundantly vascularized and innervated [3]. It is located caudal to the patella, dorsal to the patellar tendon and ventral to the femoral condyles as well as the tibial plateau. On its dorsal surface facing the joint, the IFP is covered with a synovial membrane forming an anatomical-functional unit with the knee joint [4]. This structure is connected to the intercondylar region and the synovial tube of the anterior cruciate ligament via the infrapatellar plica (lig. Mucosum), which further contributes to the perfusion of the anterior cruciate ligament through its arterial anastomosis [5]. Moreover, the IFP is attached to the anterior roots of both menisci [6]. Perfusion is provided by a periarticular arterial network, the genicular anastomosis or rete articulare genus, which is formed by branches of the popliteal artery (i.e., superior lateral and medial genicular arteries, inferior medial and lateral genicular arteries, middle genicular artery) [5]. Hence, a hyperperfused peripheral area and a relatively poor vascularized area in the center can be distinguished [5].

The IFP shows gender-dependent size differences (average volume: 21 cm^3^ in women, 29.7 cm^3^ in men). Furthermore, its size positively correlates with the patient’s BMI [7,8]. In contrast to visceral and subcutaneous adipose tissues, the IFP does not consist of storage fat, but of structural and mechanically protective building fat, and thus is subject to alimentary fluctuations to a significantly lower extent [9]. The adipocytes in the IFP for instance display a significantly smaller cell volume than in subcutaneous adipose tissue [10].

The IFP consists predominantly of lobulated white adipocytes, which are embedded in fat- and mostly collagen-type-I- and -type-III-containing connective, fibrous tissue [9,11]. Particularly the dorsal part shows a high density of sensory innervation by S100- and substance P-positive nerve endings. These originate mainly from the popliteal plexus (tibial nerve) and from the common fibular nerve as well as to a lesser extent from the femoral nerve, potentially contributing to anterior knee pain [12]. Substance P-positive nerve fibers account for about a quarter of all nerves in the IFP [12]. In general, the peripheral release of substance P in the IFP is not only responsible for the mediation of pain but also elicits a paracrine pro-inflammatory effect via secretion of IL-1β, IL-6, and IL-8 by various cell types [13,14]. This form of neurogenic inflammation is based on vasodilatation, increased vascular permeability, extravasation of plasma proteins, recruitment of immune cells, and adhesion of leukocytes with subsequent edema of the IFP, which can be visualized in imaging methods such as MRI [13,14]. In contrast to substance P, respective sensory nerve fibers secrete another neuropeptide, the calcitonin gene-related peptide (CGRP), which has a vasodilatory effect potentially leading to synovial hypertrophy or synovitis [15].

In addition to the sensory nerve fibers (substance P-positive), the IFP further contains tyrosine hydroxylase-positive sympathetic nerve fibers which exert their effect mainly via the peripheral neurotransmitter norepinephrine [16]. It is interesting to highlight here that in patients with anterior knee pain after TKA, sensory innervation predominates over sympathetic innervation in the IFP compared with conservatively treated OA patients, possibly leading to exacerbation and continuation of knee pain and local inflammation after TKA [16]. However, other studies postulate that sympathetic nerve fibers in the knee joint tend to have an opposite effect via the release of endogenous opioids, noradrenaline and inhibition of the release of substance P, thus counteracting postoperative fibrosis, restricted mobility and achieving an analgesic effect [16,17].

### 3.2. Mechanical Aspects

The posterior part of the IFP adheres to the synovial lining and exhibits structural, gap-filling and stabilizing properties in the joint cavity. The IFP fills the space between the femur, tibia and patella intraarticularly to stabilize the patella during movement and thus protect the knee joint from mechanical damage [18]. The pressure and volume of the IFP change throughout the range of motion of the knee joint. The increased tissue pressure at terminal knee flexion and extension provides stability to the patella. The IFP can also serve as a shock-absorbing cushion between the inferior patellar tendon and the anterior tibial plateau [19]. In addition, it contributes to the blood supply of the anterior cruciate ligament as well as of the patellar tendon [13]. Accordingly, its essential functions are the production and secretion of synovial fluid, promotion of sliding capacity of femoropatellar structures, and shock absorption of compressive stress as well as shearing load [5]. As described more precisely in later sections, in contrast to subcutaneous adipose tissue, the IFP contributes to mechanical stress resistance in particular with its different microscopic structures [20]. Especially during <20° and >100° knee flexion, the IFP gets mechanically stressed. This increases the risk of a particular set of pathologies such as anterior knee pain and Hoffa syndrome [5]. Chronic impingement due to hypercompression, tissue ischemia, and repetitive trauma of the IFP in the anterior compartment of the knee is considered the primary cause of Hoffa syndrome. The IFP of patients concerned presents inflammation and subsequent hypertrophy, fibrosis and impingement leading to structural degradation. A variety of symptoms is manifesting in these patients such as pain in the anterior knee joint, functional limitations and conceivably articular effusion or palpable swelling surrounding the patellar [1]. Furthermore, the IFP shows age-dependent changes even in the absence of advanced degenerated diseases. Adipocytes of the IFP increase in size with age; slightly firm collagen type I fibers increase likewise, whereas more compliant collagen type III fibers decrease, suggesting a reduction in elastic properties with cellular senescence [21].

Molecular functional aspects of the IFP and its several interactions with surrounding articular structures are described below in respective sections. Table 1 provides a summary of anatomical and functional features of the IFP.

### 3.3. Contribution to the Pathogenesis of Knee OA

Recent studies suggest that in addition to the synovial membrane, articular cartilage, and internal structures like menisci and subchondral bone, the IFP may also play a role in the development of knee OA (Figure 1) [23]. Growing scientific evidence indicates that the IFP evolved from synovial tissue considering its structure and functionality, indicating an extensive communication between the IFP and the synovia as well as the fibrous capsule [9,24]. Therefore, IFP and the synovia should be considered as an anatomical-functional unit as opposed to two separate structures that merely communicate with each other [4,25].

In the context of immunohistochemical examinations of IFPs obtained from OA patients, significant differences were described compared to healthy tissue samples: increased fibrosis, hypervascularization, enhanced lymphocyte infiltration, and increased levels of growth factors and cytokines such as vascular endothelial growth factor (VEGF) and monocyte chemotactic protein 1 (MCP-1) [11]. Figure 2 shows sections of two IFPs from OA patients resected during TKA demonstrating the different degrees of fibrosis even though the clinical and radiological diagnosed knee OA were similar. This discrepancy might depend on many factors, which have to be analyzed systematically in the future according to anthropometric data, for instance. These alterations are thought to be directly involved in molecular processes promoting knee OA by modified cytokine production. Changes in cytokine production might also affect the biomechanical properties of IFP by compromising its ability to absorb compressive and gravitational forces on the knee joint, consequently aggravating or perpetuating joint damage [11].

Biomechanical studies showed an increased compressive stiffness of IFP in OA conditions compared to healthy controls, consequently losing the ability to redistribute loads through the tissue and the surrounding biological structures. Investigations of the mechanical contribution of sub-components of the IFP on the overall mechanical behavior demonstrated a non-organized distribution of the stresses within the interlobular septa of the OA IFP compared to a homogeneous distribution of the tensile stress around the adipose lobules for the healthy IFP. Furthermore, there was an increased stiffness in the adipose lobules for OA IFP, which leads to greater overall stiffness. In addition, there is a different distribution pattern and microstructure of connective tissue in OA IFP and healthy controls. Healthy IFP shows a distribution of tensile stress along the fiber’s direction, indicating a possible capability of fibers to oppose torsional actions. In contrast, OA IFP is characterized by a circumferential distribution of tensile stresses due to a rather random distribution of fibers inside the interlobular septa [26,27].

#### 3.3.1. Secretory Profile

Adipocytes account for a vast majority of the cells in the IFP. Besides that, there are fibroblasts responsible for the production of extracellular matrix, and immune cells such as macrophages (i.e., pro-inflammatory M1 macrophages and anti-inflammatory M2 macrophages), mast cells (MC) and lymphocytes [24]. Macrophages especially are actively involved in the pathogenesis of knee OA [28]. Moreover, different tissues or cellular components of the IFP such as adipose tissue, fibroblasts, mononuclear cell infiltrations (e.g., monocytes, lymphocytes and macrophages), and MC have been shown to secrete a variety of cytokines, adipokines, and growth factors. These factors, which in turn affect cell-to-cell interactions, perpetuate the local inflammatory response, and can also induce the onset of neuropathic pain and chronic neurogenic tissue inflammation [29]. The role of these factors in OA is now discussed in more detail.

Cytokines: Immune cells in the IFP secrete a variety of cytokines. In OA patients, significantly higher levels of interleukin 6 (IL-6) and its soluble receptor (soluble interleukin-6 receptor–sIL-6R) in the IFP compared to subcutaneous adipose tissue were detected. IL-6 affects other joint components (synovium, articular cartilage, subchondral bone) through paracrine pathways and thus plays a crucial role in the progression of OA [11]. The classical IL-6 signaling pathway achieves a chondrolytic effect in knee joints via inhibition of proteoglycan synthesis in human chondrocytes. Furthermore, IL-6 suppresses the neosynthesis of collagen type II, enhances IL-1β-mediated proteoglycan degeneration, and induces the expression of various metalloproteinases (MMPs) and ADAMTS, which additionally mediate cartilage degeneration [30]. IL-1β has a strong catabolic effect on chondrocytes. Its effect is mediated via multiple pathways including upregulation of aggrecanases and MMPs, induction of anti-inflammatory mediators, and downregulation of synthesis of chondrogenic extracellular matrix [31].

Moreover, pro-inflammatory cytokines in the IFP, such as tumor necrosis factor α (TNF-α) and IL-1β, or anti-inflammatory cytokines such as IL-4, IL-10, and the hydrophobic hormone-like substance prostaglandin E2 (PGE2), and their balance can also facilitate the development of knee OA at regular concentration when compared to control groups [19].

Adipokines: Adipocytes in the IFP secrete adipokines (i.e., adiponectin, leptin, resistin, and visfatin) that contribute to inflammatory processes in the knee joint [9]. Above all, adiponectin is to be emphasized for exerting anti-inflammatory effects. Adiponectin contributes to the regulation of glucose and lipid metabolism with an anti-diabetic and anti-atherogenic effect [32]. In particular, it enhances the production of the anti-inflammatory cytokine IL-10 by M2 macrophages and binds to apoptotic cells, thereby facilitating their uptake by macrophages. The phagocytosis of apoptotic cells in turn promotes the differentiation of M2 macrophages, enabling adiponectin to protect the organism from both local and systemic inflammation [33]. Leptin, whose concentration has been shown to be increased in the serum of OA patients, stimulates chondrocytes to synthesize cartilage-degrading MMPs and pro-inflammatory cytokines [34,35]. Elevated concentrations of leptin are not only found in the serum of OA patients, but leptin is actively secreted by IFP adipocytes into the synovial fluid [24,25]. Resistin-an adipokine that is not secreted by components of the IFP but produced by monocytes and macrophages in adult individuals can lead to structural impairment of cartilaginous structures. The aforementioned adipokines also include fatty acid-binding protein 4 (FABP4), WNT1-inducible signaling pathway protein 2 (WISP2) and chemerin [19]. In this context, OA patients exhibited higher levels of pro-inflammatory adipokines such as leptin, FABP4, WISP2 and chemerin in the IFP when compared to healthy individuals [36]. FABP4 is mainly expressed in macrophages and adipocytes and facilitates the transport of free fatty acid in adipocytes for metabolic processes and therefore plays an important role in the pathogenesis of metabolic diseases [37]. Its relevance in osteoarthritis is gaining increasing evidence because an increased expression is associated with the severity of osteoarthritis [38]. WISP2 is a matricellular protein that belongs to the CCN family and plays multiple roles in a variety of pathophysiological processes, including cell proliferation, extracellular matrix and tissue regulation and regeneration, angiogenesis and fibrosis [39,40]. Chemerin, expressed in white adipose tissue, is associated with energy metabolism that facilitates glucose uptake, lipolysis, and adipocyte differentiation and it enhances the production of pro-inflammatory cytokines, as well as chondrolytic MMPs [41].

Growth factors: VEGF is a mediator of (neo)angiogenesis, playing an essential role in the etiology and development of inflammatory joint diseases, including OA. A higher expression of VEGF was demonstrated in the IFP of OA patients compared to healthy controls. VEGF levels again were positively associated with the degree of synovial vascularization, further confirming an interaction between the IFP and synovium [11].

In fact, VEGF promotes tissue damage and the genesis of pain by maintaining or enhancing the invasion of inflammatory cells and upregulating the expression of local pain receptors [42]. Basic fibroblast growth factor (bFGF), or as it is correctly termed today fibroblast growth factor 2 (FGF2), is a member of the fibroblast growth factor (FGF) family. Its role in the homeostasis of articular cartilage is controversial. FGF-2 induces proteoglycan degradation in human cartilage and inhibits sustained proteoglycan accumulation in chondrocytes [43]. Moreover, FGF-2 potently antagonizes proteoglycan synthesis, it induces the collagen-type II degrading enzyme in articular cartilage MMP-13, and promotes the expression of chondrolytic and pro-inflammatory factors such as aggrecanases (i.e., ADAMTS-5), substance P, and TNF-α-receptor [43]. In contrast, FGF-2 has an anti-inflammatory effect on human articular chondrocytes. It represses cartilage degeneration induced by IL-1 as well as the expression of chondrolytic ADAMTS-4 and ADAMTS-5 [44]. Similar observations apply to transforming growth factor β (TGF-β). TGF-β is responsible for regulating a quantity of processes including cell proliferation and tissue formation, repair and inflammation, and also maintains the differentiated chondrocyte phenotype in healthy cartilage [20]. Here, a controversial role depending on the TGF-β concentration in synovial fluid and tissue has also been described in the literature.

Table 2 summarizes cellular and molecular components present in the IFP that are involved in the development and progression of knee OA.

#### 3.3.2. Immunological Role

The previously mentioned molecular factors strongly trigger pro-inflammatory states. This results in extravasation of circulating immune cells into the IFP and synovium, whose secretion of prostaglandins as well as interleukins, in turn, promotes further extravasation of immune cells by attracting lymphocytes to the endothelium and promoting their migration into the surrounding IFP and synovium. Furthermore, the release of substance P promotes neurogenic inflammation with further vasodilation and chemotaxis [28]. The immune response, initiated by both the innate and adaptive immune systems, makes a significant contribution to the pathogenesis of OA since it is activated after cartilage damage [47,48]. Traumatic damage to the extracellular matrix (ECM) of tissues in the joint occurs due to microtraumata following continuous or repetitive overuse as well as regular aging processes [48,49]. These structural changes elicit damage-associated molecular patterns (DAMPs), which, in turn, activate the innate immune system. In particular, DAMPs initiate inflammatory processes by interacting with particle recognition particles (PRR) on the surface of immune cells (e.g., Toll-like receptors) [48].

Aside from monocytes, MC, and lymphocytes, macrophages are important cell types of the innate immune system, and they are particularly relevant for the pathogenesis and progression of OA [50]. Resident macrophages in the IFP are activated by a variety of interleukins and interferons secreted by other adipose cells within the IFP, by resident- and from blood-stream- or synovial fluid-infiltrating immune cells [25]. Those macrophages exist in heterogeneous polarization phenotypes, of which the most important and quantitively dominating are M1 and M2 macrophages. The secretion profile of macrophages depends on the respective subtype. M1 macrophages with T helper cell type 1 (Th1)-dependent differentiation facilitate inflammatory reactions through the production of pro-inflammatory cytokines (i.e., IL-1β, TNF-α, IL-6 and IL-12) and release pro-fibrotic mediators such as connective tissue growth factor (CTGF) over an extended period of time [22,24]. In this context, macrophages release catalytic enzymes as MMPs and proteoglycan-degrading aggrecanases (ADAMTS-4 and -5) become activated resulting in the degeneration of cartilage tissue [51]. In contrast, T helper cell type 2 (Th2)-dependently differentiated M2 macrophages mainly secrete IL-10, suggesting that they are anti-inflammatory trying to balance the M1 macrophage fraction [22]. Furthermore, the transcription of several peroxisome proliferator-activated receptor (PPAR)-regulated genes is upregulated in M2 macrophages, consequently increasing the expression of arginase 1 (Arg1) and IL-1 receptor antagonist (IL-1Ra) [52]. PPARγ, in turn, regulates glucose and lipid metabolism, inhibits inflammation in articular cartilage tissue, promotes differentiation of adipocytes and stimulates adiponectin secretion by activating the latter [52]. Therefore, M2 macrophages are profoundly involved in repairing damaged tissue and alleviating inflammation [52]. Furthermore, there is an upregulated expression of PPARγ genes and proteins in end-stage OA compared to early-stage OA and controls [53]. In contrast, the gene expression of PPARγ is lower in IFP than in subcutaneous adipose tissue, suggesting that a deficiency of PPARγ-driven anti-inflammatory mechanisms may contribute to the progression or pathogenesis of OA [22].

Data on the function of MC in the IFP are scarce. MC numbers are elevated in synovial tissue from OA patients compared to rheumatoid arthritis [54,55,56], and their number increases with higher Kellgren–Lawrence scores [56]. Recently, MC deficiency and pharmacologic inhibition were shown to protect against cartilage degeneration in a model for post-traumatic OA, the destabilization of the medial meniscus (DMM), while MC transfer into MC-deficient mice resulted in aggravated cartilage destruction [57]. Furthermore, inhibition of MC tryptase also preserved cartilage after DMM, indicating a deleterious role of the proteinase [57]. Indeed, it was shown previously that MC tryptase activates latent MMP-3 and -13, thereby promoting cartilage destruction in an ex vivo experiment [58]. Unlike the T cells described earlier, the presence of tissue-resident B cells in IFP apparently contributes a minor role in the development and progression of OA, because no concentration changes of these cells can be shown in IFP compared with subcutaneous adipose tissue, as well as in healthy IFP compared with OA IFP [24,25,28].

#### 3.3.3. IFP-Derived MSCs

The abundance of mesenchymal stem or stromal cells (MSCs) in form of perivascular cells within the highly vascularized IFP has resulted in the sourcing and use of MSCs derived from IFP tissues for cell-based therapies [25]. IFP-derived MSCs have recently gained popularity due to their easy accessibility compared to other stem cell sources such as bone marrow and adipose tissue, while showing similar levels of multipotentiality as well as growth and immunomodulatory potential [25]. In this context, the multipotentiality of IFP-MSCs for chondrogenic, osteogenic and adipogenic cell lines was demonstrated [29]. Given their high proliferation rate and superior chondrogenic differentiation ability, MSCs derived from the IFP can be considered a suitable source for the development of cartilage matrices to repair focal chondral defects [25].

IFP-MSC multipotentiality toward chondrogenic, osteogenic, and adipogenic lineages has been demonstrated as comparable to other MSC types with similar growth kinetics to bone marrow-derived MSC or even higher proliferation to synovial fluid MSCs. MSC differentiation capacity seems to be related to the tissue of origin. The intraarticular localization of IFP tissue with its anatomical proximity to articular cartilage provides a strong chondrogenic differentiation capacity of IFP-MSCs [25,59]. Previous studies revealed, that IFP-MSC from both healthy and knee OA patients seem to be a suitable candidate to engineer cartilaginous grafts to resurface clinically relevant defects of articular cartilage surfaces. These data suggest that the inflammatory OA environment does not inevitably prime IFP-MSC to pro-inflammatory balance and IFP-MSC still has strong properties of proliferation and multipotency [60,61]. In addition to tissue engineering using IFP-MSCs, therapeutic effects of IFP-MSCs mediated by essential functional soluble factors play a promising role. These factors are secreted by IFP-MSCs in extracellular vesicles (EV) and can be used as cell-free MSC therapy for therapeutic applications in cartilage regeneration. Here, 30–150 nm exosomes are particularly suitable due to their increased stability under multiple physiological conditions. The function of exosomes is primarily an intercellular means of communication to transfer bioactive lipids, nucleic acids or proteins from donor cells into recipient cells to achieve biological metabolic processes or contribute to cell-to-cell interaction [62,63]. The therapeutic potential of EVs for the treatment of musculoskeletal diseases has been reviewed extensively over the last years [63].

## 4. Effects of Surgical Resection of the IFP in TKA

According to today’s standard of care, the IFP is (partially) resected during most TKA procedures in order to provide optimal visualization of the operation area and easier surgical access (Figure 3) [2,64]. Currently, there is no clear recommendation as to whether the IFP should be removed or preserved during surgery due to controversial evidence regarding the clinical outcome of TKA [2,65]. A meta-analysis from Great Britain showed that 23.1% of IFPs were resected, 62.4% were partially resected and 9.8% were preserved with no difference in postoperative patient satisfaction between these groups [2]. In accordance with this low rate of IFP-preserving surgical procedures, one recent experimental animal study demonstrated that preservation of IFP may have a negative impact on the development of OA [66]. After removal of the IFP in the rodent model of guinea pigs, fibrous connective tissue developed at the site of the previously resected IFP after a couple of months. Comparing IFP-resected and IFP-preserved knees in those rodents, reduced OA-associated damage to subchondral bone and articular cartilage was observed in IFP-resected knees [66].

However, the definite consequences of IFP resection or preservation during TKA are controversially discussed and not fully understood. The following subsections provide an overview of the respective possible outcomes after IFP resection or preservation. The possible consequences of IFP resection for postoperative results are aggravated anterior knee pain [67] and increased risk of reduced blood flow in the patellar tendon with scarring and subsequent shortening of the patellar tendon [68,69]. It can be assumed that the shortening of the tendon might lead to increasingly limited mobility in the knee joint [70]. Furthermore, the complication rate might differ depending on the applied surgical technique.

### 4.1. Anterior Knee Pain

Seo et al., investigated the postoperative anterior knee pain of 448 TKAs (201 IFPs resected, 247 IFPs preserved) in the first 72 h at every 6 h in a retrospective study. They did not find any difference in postoperative pain 72 h after TKA [71]. However, maximum pain levels as measured by the numerical rating scale (NRS) 24 to 48 h after the procedure decreased significantly only in the group with resected IFP [71]. Moreover, Maculé et al. found in their OA cohort of 68 participants that patients with IFP resection experienced less pain compared to patients with preserved IFP at 1 month and 6 months post-intervention [72]. In contrast, Pinsornsak et al. found an increase in anterior knee pain with intraoperatively resected IFP after a period of 6 months after surgery in a randomized controlled trial with 90 patients [73]. Similarly, Meneghini et al. reported tendencies of enhanced pain following an intraoperative resection of the IFP in a retrospective review of 1055 TKAs [74]. Tanaka et al. compared clinical outcomes of IFP resection and synovectomy in 120 patients undergoing TKA for the treatment of OA of the knee secondary to rheumatoid arthritis as an underlying chronic disease. They observed a significantly increased incidence of anterior knee pain in patients who underwent synovectomy including IFP resection compared to patients without infrapatellar synovectomy or IFP resection [75].

Therefore, the IFP is considered a possible origin of anterior knee pain [12]. Furthermore, the IFP has been shown to contain peptide C positive and substance P positive nerve fibers [18]. In summary, the preservation of the IFP during surgery might be causative to the tendency towards increased pain during the first one to two months post-intervention. However, three to six months after surgery, the trend reverses, with anterior knee pain occurring more frequently in the group with IFP resection [76,77].

### 4.2. Vascularization of the Patella

The postoperative quality of blood flow to the kneecap is being discussed in an equally controversial manner. Reduced patellar perfusion potentially resulting in avascular necrosis of the patella has been described as a consequence of IFP resections [78]. Subramanyam et al. demonstrated that a subtotal IFP resection (resection of a very large part of the IFP) significantly impedes the vascularization of the patella, whereas a partial excision (resection of 50% of the IFP or less) does not restrict perfusion [78]. In contrast, McMahon et al. did not observe circulatory disorders in the patella as a result of IFP resection one month post-surgery [79].

### 4.3. Range of Motion and Functional Outcome

Resection of the IFP during TKA does neither result in significant differences in Knee Society Scores (KSS) and functional subscores nor in a reduced extent of knee flexion capacity, as demonstrated by Pinsornsak et al., in a prospective study [73]. Similarly, Meneghini et al. observed no significant differences with regard to the postoperative range of motion or functional scores comparing patients with and without IFP resection [74]. In contrast, Tanaka et al. found that resection of the IFP results in greater restriction of movements (especially knee flexion), a slight weakness of the quadriceps muscles and a shortening of the patellar tendon diagnosed via radiological imaging [75]. However, this study exclusively investigates patients with knee OA secondary to rheumatoid arthritis as opposed to patients with primary OA.

### 4.4. Length of the Patellar Tendon

The existing literature presents conflicting statements regarding the postoperative, radiological change in length of the patellar tendon. In a prospective study by Pinsornsak et al., the resection of the IFP during TKA did not result in a significant difference in the patellar tendon length after a follow-up period of up to twelve months after surgery [73]. Maculé et al. were in accordance with these findings investigating similar data points during a different observation period (three and six months postoperatively) using a prospective study design [72]. Furthermore, İmren et al. did not observe a shortening of the patellar tendon five years after the respective procedure [80]. In contrast, Lemon et al. reported a significant shortening of the patellar tendon in IFP-resected patients one to three years after TKA [68] as did Tanaka et al. [75]. Similarly, Chougule et al. reported a significant shortening of the patellar tendon one and five years after IFP resection [69].

### 4.5. Complications

Few data exist on complications associated with the resection of the IFP compared to the preservation of the IFP. Seo et al. found that despite the longer surgery time, preservation of the IFP is associated with significantly reduced rates of postoperative wound healing disorders [71]. In this context, a wound-healing disorder was defined as prolonged wound secretion for more than three days post-intervention. Notably, two wound revisions were performed during the first eight weeks postoperatively in the IFP resection group only [71]. In a study by Pinsornsak et al., no complications were documented following either IFP resection or IFP preservation [73].

## 5. Conclusions

In summary, it is reasonable to postulate that with the growing number of endoprosthetic interventions on the knee joint due to OA, the focus shifts to preventing insufficient postoperative results. Therefore, the present article outlined the anatomical principles, the role in the development of OA of the knee and the consequences of resection or preservation of the IFP in the context of TKA. The aspects presented in this review might contribute to or improve guidelines for TKA procedures and help standardize surgical decision-making among practitioners. While there were no differences in KSS after IFP resection, the effects of resection on patellar tendon length, postoperative knee pain, and the range of motion varied across studies, sometimes presenting contradicting outcomes (Table 3). Yet, there is still no clear evidence that surgeons should be advised to preserve the IFP in the absence of macroscopic fibrosis, if a sufficient intraoperative overview can be achieved without (partial) resection. Further prospective randomized controlled studies and clinical trials are needed from this preliminary point of view before suggesting the optimal surgical technique during endoprosthetic operations for a new standard of care in OA.

## Figures and Tables

**Figure 1 biomedicines-10-01071-f001:**
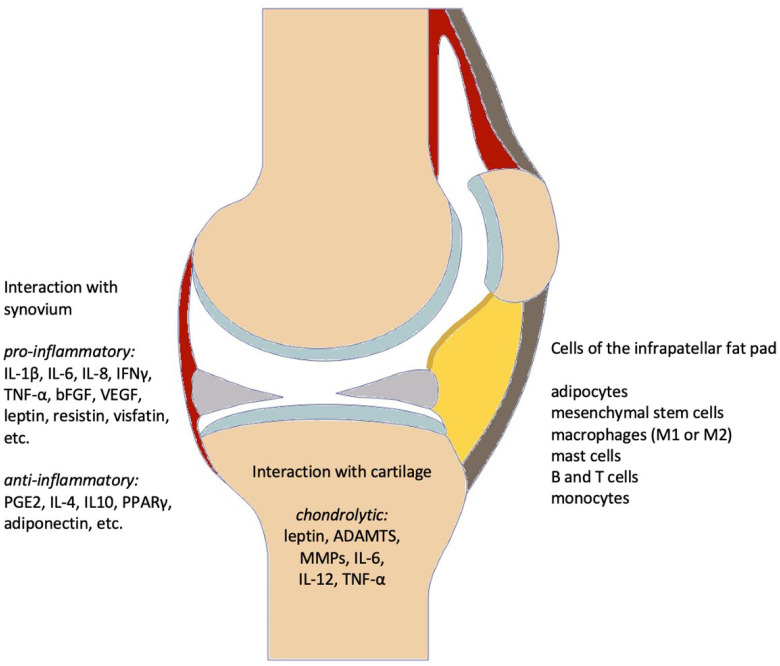
Mediators released by different cell types of the IFP and their molecular interactions with neighboring articular cartilage and synovium relevant in knee OA. brown: tibia, femur and patella, yellow: IFP, orange: synovial lining of the dorsal part of the IFP, red: joint capsule, grey: menisci, light blue: articular cartilage.

**Figure 2 biomedicines-10-01071-f002:**
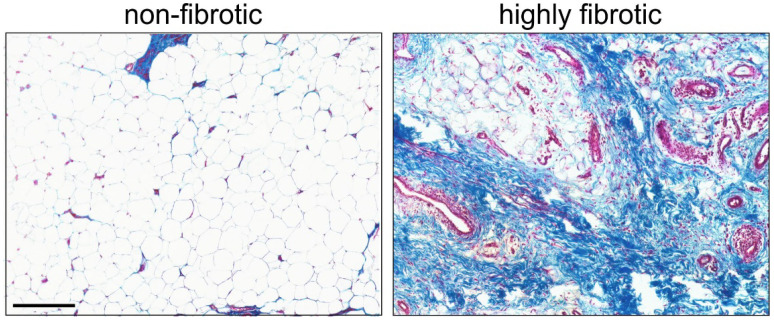
Masson trichrome staining of a non-fibrotic IFP (**left**) and a highly fibrotic IFP (**right**) of patients undergoing TKA for primary knee OA. In the left image, there is plenty of adipocytes and low fibrosis, whereas the right image shows fibrotic tissue of IFP with a reduction of adipocytes and a great increase of collagen fibers as well as intense neovascularization. Collagen fibers stained blue, nuclei stained dark purple and cytoplasm/muscle fibers stained red. Bar: 200 μm.

**Figure 3 biomedicines-10-01071-f003:**
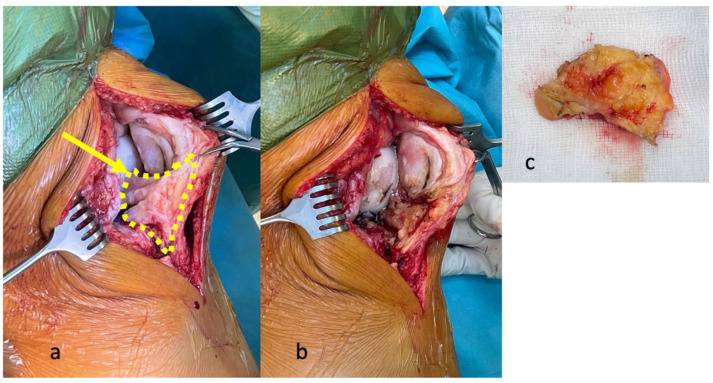
Intraoperative visualization of the IFP during TKA: (**a**) intraoperative view of the IFP during implantation of a TKA (yellow dotted line) with infrapatellar plica (lig. mucosum), (**b**) significant improvement of the intraoperative overview after resection of the IFP, (**c**) view on the resected IFP with its dorsal synovial covering, IFP–infrapatellar fat pad.

**Table 1 biomedicines-10-01071-t001:** Summary of the anatomical and functional features of the IFP [1,3,4,5,6,9,11,12,22].

Feature	Findings
Anatomy	*macroscopic:* location: intraarticular, extrasynovial [3,6] infrapatellar plica/lig. mucosum (vascularization of anterior cruciate ligament) [5] *microscopic:* hypovascularized zone in the center of univacuolar lobulated adipose tissue [5] with septa of collagenous connective tissue [9,11] covered with synovium on the dorsal surface [4] hypervascularized and strongly innervated zones in the periphery [3,5]
Vascularisation	genicular anastomosis (rete articulare genus) [5] branches of the popliteal artery [5]: ○ superior medial and lateral genicular artery ○ inferior medial and lateral genicular artery ○ middle genicular artery
Innervation	substance P- and S100-positive nerve endings [11,12] popliteal plexus [11]: ○ tibial nerve ○ posterior branch of obturator nerve common fibular nerve [11] femoral nerve [11]
Function	*mechanic:* production and secretion of synovial fluid [5] shock absorption of pressure and shear loads [1,5] *molecular:* secretion of pro- and anti-inflammatory cytokines, adipokines, growth factors [12] endocrine, autocrine and paracrine interaction with synovium, articular cartilage and subchondral bone [3,12,13] immunological balance between M1 and M2 macrophages [22]

**Table 2 biomedicines-10-01071-t002:** Expression of inflammatory factors and presence of mononuclear cells in the human IFP during OA pathogenesis and progression.

Components	Pro-Inflammatory	Anti-Inflammatory
Cytokines/Mediators	IL-α [13,20] IL-1β [13,20,25] sIL-6R [13,19,20] TNF-α [13,19,20,25,45] IFNγ [13,19,25,45] IL-6 [13,19,20,25,45,46] IL-8 [13,19,20,25] IL-12 [19,25] IL-15 [20] IL-17 [45] IL-18 [20] IL-33 [20] IL-36α [20] IL-36β [20] IL-36γ [20]	IL-1Ra [19,20] IL-4 [19] IL-10 [19,45] IL-36Ra [20] IL-37 [20] IL-38 [20] Arg1 [19] PGE2 [19,20] PGF2α [25]
Adipokines	resistin [19,45] leptin [13,19,45,46] visfatin [13,19,45] FABP4 [19,46] WISP2 [19] chemerin [19]	adiponectin [13,19,45,46]
Growth factors	VEGF [13,19,20,46] FGF-2 [13] TGF-β [19,20,46]	FGF-2 [13] TGF-β [19,20,46]
Gene expression		PPARγ [19,46]
Immune cells	M1-macrophages [13,19,25] CD4+ T-cells (Th1) [19,25] Mast cells [13,25,45]	M2-macrophages [13,19,25] CD8+ T-cells (Th2) [19,25]

Abbreviations: IL—interleukin, sIL-6R—soluable interleukin 6 Receptor, TNFα—tumor necrosis factor α, IFNγ—interferon γ, FABP4—fatty acid-binding protein 4, WISP2—WNT1-inducible signaling pathway protein 2, VEGF—vascular endothelial growth factor, Th1—T1-helper cells, Th2—T2-helper cells, CD—cluster of differentiation, Ra—receptor agonist, TGF-β—transforming growth factor β, Arg1—arginase 1, FGF-2—fibroblast growth factor 2, PGE2—prostaglandin E_2_, PGF2α—prostaglandin F_2α_, PPARγ—peroxisome-proliferator-activated receptor γ.

**Table 3 biomedicines-10-01071-t003:** Summary of the clinical consequences of IFP resection or preservation during TKA from the studies included for analysis in this review.

Study	Primary Outcome	Results
Chougule et al. [69]	Length of patellar tendon	Shortening of patellar tendon after IFP-resection
İmren et al. [80]	Range of motion	No difference
	Length of patellar tendon	No difference
Lemon et al. [68]	Length of patellar tendon	Shortening of patellar tendon after IFP-resection
Maculé et al. [72]	Anterior knee pain	Decreased pain after IFP-resection
	Range of motion	No difference
	Length of patellar tendon	No difference
McMahon et al. [79]	Vascularization of the patella	No difference
Meneghini et al. [74]	Anterior knee pain	Increased pain after IFP-resection
	Range of motion	No difference
	Length of patellar tendon	No difference
	Insall–Salvati ratio	No difference
	Knee society score	No difference
Pinsornsak et al. [73]	Anterior knee pain	Increased pain after IFP-resection
	Range of motion	No difference
	Length of patellar tendon	No difference
	Insall–Salvati ratio	No difference
	Knee society score	No difference
Seo et al. [71]	Anterior knee pain	No difference
Subramanyam et al. [78]	Vascularization of the patella	Hypoperfusion of patella after IFP-resection
Tanaka et al. [75]	Anterior knee pain	Increased pain after IFP-resection
	Range of motion	Decreased after IFP-resection
	Length of patellar tendon	Shortening of patellar tendon after IFP-resection
	Insall–Salvati ratio	Shortening of patellar tendon after IFP-resection
Chougule et al. [69]	Length of patellar tendon	Shortening of patellar tendon after IFP-resection
İmren et al. [80]	Range of motion	No difference
	Length of patellar tendon	No difference
Lemon et al. [68]	Length of patellar tendon	Shortening of patellar tendon after IFP-resection
Maculé et al. [72]	Anterior knee pain	Decreased pain after IFP-resection
	Range of motion	No difference
	Length of patellar tendon	No difference
McMahon et al. [79]	Vascularization of the patella	No difference
Meneghini et al. [74]	Anterior knee pain	Increased pain after IFP-resection
	Range of motion	No difference
	Length of patellar tendon	No difference
	Insall–Salvati ratio	No difference
	Knee society score	No difference
Pinsornsak et al. [73]	Anterior knee pain	Increased pain after IFP-resection
	Range of motion	No difference
	Length of patellar tendon	No difference
	Insall–Salvati ratio	No difference
	Knee society score	No difference
Seo et al. [71]	Anterior knee pain	No difference
Subramanyam et al. [78]	Vascularization of the patella	Hypoperfusion of patella after IFP-resection
Tanaka et al. [75]	Anterior knee pain	Increased pain after IFP-resection
	Range of motion	Decreased after IFP-resection
	Length of patellar tendon	Shortening of patellar tendon after IFP-resection
	Insall–Salvati ratio	Shortening of patellar tendon after IFP-resection

## Data Availability

Not applicable.

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
