# Peer review of "The Corpus Adiposum Infrapatellare (Hoffa’s Fat Pad)—The Role of the Infrapatellar Fat Pad in Osteoarthritis Pathogenesis"

_biomedicines, 2022, doi:10.3390/biomedicines10051071_

Round 1

Reviewer 1 Report

The paper aims at defining the role of the infrapatellar fat pad in osteoarthritis pathogenesis. Preliminarily, anatomical and physiological information is provided. The paper is well written and all the dfferent topics are well described.

Some minor comments can be reported:

Page 2: “IFP does not consist of storage fat, but of structural and mechanically protective building fat”, the authors are encouraged to extend this topic. The mechanical functionality of the Hoffa’s fat pad is relevant and should be discussed. In details, comments about the relationship between micro- and meso-structural configuration are expected.

The paper looks more being a Review.

Author Response

Manuscript ID: biomedicines-1700669

Dear Editors,

It is our great pleasure to submit the revised version of our manuscript (biomedicines-1700669, review article) entitled

“The corpus adiposum infrapatellare (Hoffa's fat pad) - the role of the infrapatellar fat pad in osteoarthritis pathogenesis”

written by Frank Zaucke, Marco Brenneis, Anna E. Rapp, Patrizia Pollinger, Rebecca Sohn, Zsuzsa Jenei-Lanzl, Andrea Meurer and by myself.

We thank the Reviewers for their valuable comments. Based on their recommendations as well as on their constructive critique, we were able to substantially improve our manuscript. As suggested, we provide additional text on missing aspects and corrected all minor mistakes.

Our point-by-point responses to all comments and suggestions are detailed in the following Rebuttal Letter referring to the marked copy of the manuscript.

I would like to express my thanks also on behalf of the other authors. We are very pleased that our review article was found interesting and that the review process went so quickly.

We hope that we have satisfactorily addressed all concerns and that the revised version will now be found suitable for publication in Biomedicines.

Looking forward to hearing from you.

Sincerely yours

Dr. Sebastian Braun, MD (Corresponding author)

Universal Hospital Frankfurt

Department of Orthopedics (Friedrichsheim)

University Hospital Frankfurt

Marienburgstraße 2

60528 Frankfurt/Main, Germany

Phone: +49 (0) 69 6301 941704

Email: sebastian.braun@kgu.de; s.b.braun@gmx.de

Rebuttal Letter

Review Report Round 1 (Reviewer 1):

Page 2: “IFP does not consist of storage fat, but of structural and mechanically protective building fat”, the authors are encouraged to extend this topic. The mechanical functionality of the Hoffa’s fat pad is relevant and should be discussed. In details, comments about the relationship between micro- and meso-structural configuration are expected.

Our Response:

Thank you for your review and your comments. The mechanical functionality and biomechanical behavior of the microstructural configuration of Hoffa’s fat pad has been added and is further discussed in a section below. (Please see “3.2. Mechanical functional aspects” and “3.3. Contribution to the pathogenesis of knee OA”

Review Report Round 1 (Reviewer 2):

  1. It would be important to add a brief section of methods used for this review (database used, criteria of selection, years considered, languages etc).

Our Response: Thank you for your comment. We added a brief description of how the literature search was performed – Lines 41-48.

  1. In the section “functional aspects” is very short. It should be improved.

Our Response: Thank you for your opinion on this section. We further elaborated on functional aspects and extended and improved this section. Lines 97-126.

  1. There is no mention about the age-dependent remodeling of the IFP.

Our Response: Thank you for this advice. We now mention the age-dependent remodeling of the IFP – Lines 119-123.

  1. The caption of the figures should be added after the figure and not before.

Our Response: We changed this accordingly.

  1. Figures should be reported where they are cited and not before.

Our Response: This was changed and figures were moved accordingly.

  1. Lines 109-110: the authors did not mention the meniscus.

Our Response: We now mention the menisci as internal structures. Thank you mentioning our missing reference. This was implemented – Line 136.

  1. Line 117: “can be were described” should be corrected.

Our Response: This was corrected.

  1. Lines 120-124: the authors discuss about the possibility that Changes in cytokine production might also affect the biomechanical properties of IFP by compromising its ability to absorb compressive and gravitational forces on the knee joint. To this regard, a recent paper on the characterization of the IFP biomechanical behaviour has been published and should be discussed.

Our Response: Thank you for pointing out the importance of the IFP mechanical behaviour. We agree that it is important to address this issue and now describe and discuss the biomechanical behavior of IFP in more detail. We added a recent paper as a reference – Lines 162-174.

  1. Figure 2: There is a big difference between the two OA patients reported by the authors. Why? Is the Kellgren Lawrence of these patients comparable? Did the patients have comparable BMI?

Our Response: We did not intend to go into much detail with these two histologic images, but only aimed to show that fibrosis can occur in the IFP during OA. Both samples were from patients that underwent TKA because of their advanced knee OA. Therefore, there is no clear correlation between radiological grade according to Kellgren-Lawrence and degree of fibrosis. Certainly, it will be very interesting to investigate the relationship between the degree of fibrosis and various anthropomorphologic parameters in the future. We explained this in the revised version. – Lines 166-170.

  1. Line 130: “Masson-Trichrom” should be “Masson-Trichrome”.

Our Response: This was corrected.

  1. Lines 143-144: “Moreover, different tissues of the IFP have been shown to secrete a variety of cytokines, adipokines, and growth factors.”. What are the different tissues of the IFP?

Our Response: We provided more clarity, which cells and components are involved and contribute to growth factor secretion. We added this information. – Lines 196-197.

  1. Lines 173-176: the authors discuss about leptin in serum. What about leptin and IFP?

Our Response: Thank you for drawing our attention to a missing reference. We added a sentence describing leptin secretion in the IPF – Lines 230-232.

  1. Lines 179-181: it is unclear the link of this part with adipokines.

Our Response: We moved these lines and hope that the link is now clearer. – Lines 313-315.

  1. Lines 181-183: could the authors expand this part?

Our Response: We expanded this part and added a new paragraph – Lines 241-251.

  1. In general, a brief introduction of the proteins discussed, especially the little-known ones, would be useful.

Our Response: Thanks for this important comment. We added a brief introduction as suggested – Lines 241-251.

  1. Regarding growth factors, the authors reported VEGF, FGF-2 and bFGF in table 1. However, only VEGF was discussed in the growth factor section (lines 186-193). The authors should add and discuss also the other growth factors.

Our Response: We agree with you that the discussion on other growth falls short. We added information on other growth factors in a new paragraph – Lines 259-273.

  1. Table 1 should be improved. References for each pro-inflammatory and anti-inflammatory protein should be added. It would be important also to add more details (for example, if the data were obtained in humans or cells or animal models, if the protein levels increase/decrease etc). Regarding immune cells in IFP, mast cells and B cells are missing.

Our Response: Thank you for pointing this out. We have adjusted the title of the table according to your recommendations. All data reported are from human cells. For the sake of clarity, the literature references that make up this summary table are indicated in the heading. The respective factors indicated in the columns are in each case either pro- or anti-inflammatory. When they are mentioned, the expression is meant. The severity of osteoarthritis correlates with increased expression in most cases according to those studies. Information about mast cells and B-cells was added – Lines 330-342.

  1. Lines 215-217: references should be added.

Our Response: References were added.

  1. Lines 217-219: the sentence is unclear. References should be added.

Our Response: The sentence was revised. – Lines 302-306.

  1. Lines 226-231: references are missing. This part needs to be improved. For example, it is unclear if PPARgamma is produced by IFP.

Our Response: Thank you for your comment. We explained the impact of PPARgamma and added a sentence.  Lines 324-328

  1. The immunological role of IFP should be improved.

Our Response: Thank you for this helpful comment. The paragraph was revised substantially and important references to mononuclear cells were added.  – Lines 285-342.

  1. Table 2: the authors should use the layout of the journal. References should be added.

Our Response: We now use the laypout of the journal and references were added.

  1. It is unclear why table 2, which is a summary of anatomical and functional features of IFP, is cited in the section related to the immunological role of IFP. It should be moved.

Our Response: In the revised version, the table was moved to section 3.2.

  1. Section “IFP-derived MSCs”: this section needs to improved as there is confusion. In particular, it is unclear if the authors refer to OA IFP-derived MSCs or healthy IFP-derived MSCs. This is important as it seems that OA IFP-derived MSCs might be primed by the inflammatory OA environment. There is no mention on the possible role of exosome derived from IFP-MSCs. Are there clinical studies/trials based on the use of these stem cells in OA?

Our Response: Thank you for your comment. We revised and thereby improved this part. – Lines 359-373. We agree with the reviewer that IFP MSCs play an important role and might be used in therapeutic approaches. We made some additions but did not want to go in further detail as extensive reviews on this topic have been published recently. References to these reviews were added: Liao, H.J., et al. Potential of Using Infrapatellar-Fat-Pad-Derived Mesenchymal Stem Cells for Therapy in Degenerative Arthritis: Chondrogenesis, Exosomes, and Transcription Regulation. Biomolecules 2022,

Herrmann, M., et al. Extracellular Vesicles in Musculoskeletal Pathologies and Regeneration. Front Bioeng Biotechnol 2020.

  1. Lines 255-258: references are absent.

Our Response: References were added.

  1. Line 261-262: the authors reported that one study demonstrated that preservation of IFP may have a negative impact on the development of OA. Could the authors cite this study?

Our Response: The study demonstrating that preservation of IFP may have a negative impact on the development of OA was cited: Afzali, M.F, et al. Early removal of the infrapatellar fat pad beneficially alters the pathogenesis of moderate stage idiopathic knee osteoarthritis in the Dunkin Hartley guinea pig. bioRxiv 2022, – Line 420.

  1. “Effects of surgical resection of the IFP in total knee arthroplasty”: in the first part of this section, the authors discuss about IFP resection during knee surgery. Then, they divided the section in subsections: anterior knee pain etc. The link between the subsection is unclear. It should be explained in the first part of the section. Since it is still under debate the removal of IFP, a table reporting the studies pro and cons this point would be useful for the reader. It would be useful to add a paragraph regarding clinical trials.

Our Response: We explained why we divided the following section into those subjections. The following findings are the possible clinical outcomes after IFP resection or preservation during TKA of the knee (Lines: 389-391) and the respective studies are summarized in an additional table (table 3).

  1. The authors are encouraged to add number to each section and subsection using the guidelines of the journal.

Our Response: We numbered sections and subsections as suggested.

  1. Abbreviations should be used consistently throughout the manuscript (for example, OA).

Our Response: We revised the use of abbreviations.

  1. Reference should be checked and formatted using the guidelines of the journal.

Our Response: We checked and formatted the references according to the journal guidelines.

Reviewer 2 Report

Manuscript entitled: “The corpus adiposum infrapatellare (Hoffa's fat pad) - the role of the infrapatellar fat pad in osteoarthritis pathogenesis”.

The review is well-written. However, there are several points that should be addressed by the authors to improve the review.

  1. It would be important to add a brief section of methods used for this review (database used, criteria of selection, years considered, languages etc).
  2. In the section “functional aspects” is very short. It should be improved.
  3. There is no mention about the age-dependent remodeling of the IFP.
  4. The caption of the figures should be added after the figure and not before.
  5. Figures should be reported where they are cited and not before.
  6. Lines 109-110: the authors did not mention the meniscus.
  7. Line 117: “can be were described” should be corrected.
  8. Lines 120-124: the authors discuss about the possibility that Changes in cytokine production might also affect the biomechanical properties of IFP by compromising its ability to absorb compressive and gravitational forces on the knee joint. To this regard, a recent paper on the characterization of the IFP biomechanical behaviour has been published and should be discussed.
  9. Figure 2: There is a big difference between the two OA patients reported by the authors. Why? Is the Kellgren Lawrence of these patients comparable? Did the patients have comparable BMI?
  10. Line 130: “Masson-Trichrom” should be “Masson-Trichrome”.
  11. Lines 143-144: “Moreover, different tissues of the IFP have been shown to secrete a variety of cytokines, adipokines, and growth factors.”. What are the different tissues of the IFP?
  12. Lines 173-176: the authors discuss about leptin in serum. What about leptin and IFP?
  13. Lines 179-181: it is unclear the link of this part with adipokines.
  14. Lines 181-183: could the authors expand this part?
  15. In general, a brief introduction of the proteins discussed, especially the little known ones, would be useful.
  16. Regarding growth factors, the authors reported VEGF, FGF-2 and bFGF in table 1. However, only VEGF was discussed in the growth factor section (lines 186-193). The authors should add and discuss also the other growth factors.
  17. Table 1 should be improved. References for each pro-inflammatory and anti-inflammatory protein should be added. It would be important also to add more details (for example, if the data were obtained in humans or cells or animal models, if the protein levels increase/decrease etc). Regarding immune cells in IFP, mast cells and B cells are missing.
  18. Lines 215-217: references should be added.
  19. Lines 217-219: the sentence is unclear. References should be added.
  20. Lines 226-231: references are missing. This part needs to be improved. For example, it is unclear if PPARgamma is produced by IFP.
  21. The immunological role of IFP should be improved.
  22. Table 2: the authors should use the layout of the journal. References should be added.
  23. It is unclear why table 2, which is a summary of anatomical and functional features of IFP, is cited in the section related to the immunological role of IFP. It should be moved.
  24. Section “IFP-derived MSCs”: this section needs to improved as there is confusion. In particular, it is unclear if the authors refer to OA IFP-derived MSCs or healthy IFP-derived MSCs. This is important as it seems that OA IFP-derived MSCs might be primed by the inflammatory OA environment. There is no mention on the possible role of exosome derived from IFP-MSCs. Are there clinical studies/trials based on the use of these stem cells in OA?
  25. Lines 255-258: references are absent.
  26. Line 261-262: the authors reported that one study demonstrated that preservation of IFP may have a negative impact on the development of OA. Could the authors cite this study?
  27. “Effects of surgical resection of the IFP in total knee arthroplasty”: in the first part of this section, the authors discuss about IFP resection during knee surgery. Then, they divided the section in subsections: anterior knee pain etc. The link between the subsection is unclear. It should be explained in the first part of the section. Since it is still under debate the removal of IFP, a table reporting the studies pro and cons this point would be useful for the reader. It would be useful to add a paragraph regarding clinical trials.
  28. The authors are encouraged to add number to each section and subsection using the guidelines of the journal.
  29. Abbreviations should be used consistently throughout the manuscript (for example, OA).
  30. Reference should be checked and formatted using the guidelines of the journal.

Author Response

Manuscript ID: biomedicines-1700669

Dear Editors,

It is our great pleasure to submit the revised version of our manuscript (biomedicines-1700669, review article) entitled

“The corpus adiposum infrapatellare (Hoffa's fat pad) - the role of the infrapatellar fat pad in osteoarthritis pathogenesis”

written by Frank Zaucke, Marco Brenneis, Anna E. Rapp, Patrizia Pollinger, Rebecca Sohn, Zsuzsa Jenei-Lanzl, Andrea Meurer and by myself.

We thank the Reviewers for their valuable comments. Based on their recommendations as well as on their constructive critique, we were able to substantially improve our manuscript. As suggested, we provide additional text on missing aspects and corrected all minor mistakes.

Our point-by-point responses to all comments and suggestions are detailed in the following Respond Letter referring to the marked copy of the manuscript.

I would like to express my thanks also on behalf of the other authors. We are very pleased that our review article was found interesting and that the review process went so quickly.

We hope that we have satisfactorily addressed all concerns and that the revised version will now be found suitable for publication in Biomedicines.

Looking forward to hearing from you.

Sincerely yours

Dr. Sebastian Braun, MD (Corresponding author)

Universal Hospital Frankfurt

Department of Orthopedics (Friedrichsheim)

University Hospital Frankfurt

Marienburgstraße 2

60528 Frankfurt/Main, Germany

Phone: +49 (0) 69 6301 941704

Email: sebastian.braun@kgu.de; s.b.braun@gmx.de

Respond Letter

Review Report Round 1 (Reviewer 1):

Page 2: “IFP does not consist of storage fat, but of structural and mechanically protective building fat”, the authors are encouraged to extend this topic. The mechanical functionality of the Hoffa’s fat pad is relevant and should be discussed. In details, comments about the relationship between micro- and meso-structural configuration are expected.

Our Response:

Thank you for your review and your comments. The mechanical functionality and biomechanical behavior of the microstructural configuration of Hoffa’s fat pad has been added and is further discussed in a section below. (Please see “3.2. Mechanical functional aspects” and “3.3. Contribution to the pathogenesis of knee OA”

Review Report Round 1 (Reviewer 2):

  1. It would be important to add a brief section of methods used for this review (database used, criteria of selection, years considered, languages etc).

Our Response: Thank you for your comment. We added a brief description of how the literature search was performed – Lines 41-48.

  1. In the section “functional aspects” is very short. It should be improved.

Our Response: Thank you for your opinion on this section. We further elaborated on functional aspects and extended and improved this section. Lines 97-126.

  1. There is no mention about the age-dependent remodeling of the IFP.

Our Response: Thank you for this advice. We now mention the age-dependent remodeling of the IFP – Lines 119-123.

  1. The caption of the figures should be added after the figure and not before.

Our Response: We changed this accordingly.

  1. Figures should be reported where they are cited and not before.

Our Response: This was changed and figures were moved accordingly.

  1. Lines 109-110: the authors did not mention the meniscus.

Our Response: We now mention the menisci as internal structures. Thank you mentioning our missing reference. This was implemented – Line 136.

  1. Line 117: “can be were described” should be corrected.

Our Response: This was corrected.

  1. Lines 120-124: the authors discuss about the possibility that Changes in cytokine production might also affect the biomechanical properties of IFP by compromising its ability to absorb compressive and gravitational forces on the knee joint. To this regard, a recent paper on the characterization of the IFP biomechanical behaviour has been published and should be discussed.

Our Response: Thank you for pointing out the importance of the IFP mechanical behaviour. We agree that it is important to address this issue and now describe and discuss the biomechanical behavior of IFP in more detail. We added a recent paper as a reference – Lines 162-174.

  1. Figure 2: There is a big difference between the two OA patients reported by the authors. Why? Is the Kellgren Lawrence of these patients comparable? Did the patients have comparable BMI?

Our Response: We did not intend to go into much detail with these two histologic images, but only aimed to show that fibrosis can occur in the IFP during OA. Both samples were from patients that underwent TKA because of their advanced knee OA. Therefore, there is no clear correlation between radiological grade according to Kellgren-Lawrence and degree of fibrosis. Certainly, it will be very interesting to investigate the relationship between the degree of fibrosis and various anthropomorphologic parameters in the future. We explained this in the revised version. – Lines 166-170.

  1. Line 130: “Masson-Trichrom” should be “Masson-Trichrome”.

Our Response: This was corrected.

  1. Lines 143-144: “Moreover, different tissues of the IFP have been shown to secrete a variety of cytokines, adipokines, and growth factors.”. What are the different tissues of the IFP?

Our Response: We provided more clarity, which cells and components are involved and contribute to growth factor secretion. We added this information. – Lines 196-197.

  1. Lines 173-176: the authors discuss about leptin in serum. What about leptin and IFP?

Our Response: Thank you for drawing our attention to a missing reference. We added a sentence describing leptin secretion in the IPF – Lines 230-232.

  1. Lines 179-181: it is unclear the link of this part with adipokines.

Our Response: We moved these lines and hope that the link is now clearer. – Lines 313-315.

  1. Lines 181-183: could the authors expand this part?

Our Response: We expanded this part and added a new paragraph – Lines 241-251.

  1. In general, a brief introduction of the proteins discussed, especially the little-known ones, would be useful.

Our Response: Thanks for this important comment. We added a brief introduction as suggested – Lines 241-251.

  1. Regarding growth factors, the authors reported VEGF, FGF-2 and bFGF in table 1. However, only VEGF was discussed in the growth factor section (lines 186-193). The authors should add and discuss also the other growth factors.

Our Response: We agree with you that the discussion on other growth falls short. We added information on other growth factors in a new paragraph – Lines 259-273.

  1. Table 1 should be improved. References for each pro-inflammatory and anti-inflammatory protein should be added. It would be important also to add more details (for example, if the data were obtained in humans or cells or animal models, if the protein levels increase/decrease etc). Regarding immune cells in IFP, mast cells and B cells are missing.

Our Response: Thank you for pointing this out. We have adjusted the title of the table according to your recommendations. All data reported are from human cells. For the sake of clarity, the literature references that make up this summary table are indicated in the heading. The respective factors indicated in the columns are in each case either pro- or anti-inflammatory. When they are mentioned, the expression is meant. The severity of osteoarthritis correlates with increased expression in most cases according to those studies. Information about mast cells and B-cells was added – Lines 330-342.

  1. Lines 215-217: references should be added.

Our Response: References were added.

  1. Lines 217-219: the sentence is unclear. References should be added.

Our Response: The sentence was revised. – Lines 302-306.

  1. Lines 226-231: references are missing. This part needs to be improved. For example, it is unclear if PPARgamma is produced by IFP.

Our Response: Thank you for your comment. We explained the impact of PPARgamma and added a sentence.  Lines 324-328

  1. The immunological role of IFP should be improved.

Our Response: Thank you for this helpful comment. The paragraph was revised substantially and important references to mononuclear cells were added.  – Lines 285-342.

  1. Table 2: the authors should use the layout of the journal. References should be added.

Our Response: We now use the laypout of the journal and references were added.

  1. It is unclear why table 2, which is a summary of anatomical and functional features of IFP, is cited in the section related to the immunological role of IFP. It should be moved.

Our Response: In the revised version, the table was moved to section 3.2.

  1. Section “IFP-derived MSCs”: this section needs to improved as there is confusion. In particular, it is unclear if the authors refer to OA IFP-derived MSCs or healthy IFP-derived MSCs. This is important as it seems that OA IFP-derived MSCs might be primed by the inflammatory OA environment. There is no mention on the possible role of exosome derived from IFP-MSCs. Are there clinical studies/trials based on the use of these stem cells in OA?

Our Response: Thank you for your comment. We revised and thereby improved this part. – Lines 359-373. We agree with the reviewer that IFP MSCs play an important role and might be used in therapeutic approaches. We made some additions but did not want to go in further detail as extensive reviews on this topic have been published recently. References to these reviews were added: Liao, H.J., et al. Potential of Using Infrapatellar-Fat-Pad-Derived Mesenchymal Stem Cells for Therapy in Degenerative Arthritis: Chondrogenesis, Exosomes, and Transcription Regulation. Biomolecules 2022,

Herrmann, M., et al. Extracellular Vesicles in Musculoskeletal Pathologies and Regeneration. Front Bioeng Biotechnol 2020.

  1. Lines 255-258: references are absent.

Our Response: References were added.

  1. Line 261-262: the authors reported that one study demonstrated that preservation of IFP may have a negative impact on the development of OA. Could the authors cite this study?

Our Response: The study demonstrating that preservation of IFP may have a negative impact on the development of OA was cited: Afzali, M.F, et al. Early removal of the infrapatellar fat pad beneficially alters the pathogenesis of moderate stage idiopathic knee osteoarthritis in the Dunkin Hartley guinea pig. bioRxiv 2022, – Line 420.

  1. “Effects of surgical resection of the IFP in total knee arthroplasty”: in the first part of this section, the authors discuss about IFP resection during knee surgery. Then, they divided the section in subsections: anterior knee pain etc. The link between the subsection is unclear. It should be explained in the first part of the section. Since it is still under debate the removal of IFP, a table reporting the studies pro and cons this point would be useful for the reader. It would be useful to add a paragraph regarding clinical trials.

Our Response: We explained why we divided the following section into those subjections. The following findings are the possible clinical outcomes after IFP resection or preservation during TKA of the knee (Lines: 389-391) and the respective studies are summarized in an additional table (table 3).

  1. The authors are encouraged to add number to each section and subsection using the guidelines of the journal.

Our Response: We numbered sections and subsections as suggested.

  1. Abbreviations should be used consistently throughout the manuscript (for example, OA).

Our Response: We revised the use of abbreviations.

  1. Reference should be checked and formatted using the guidelines of the journal.

Our Response: We checked and formatted the references according to the journal guidelines.

Round 2

Reviewer 2 Report

The review improved after the  revision.

No addional comments.

I have only noticed that table 3 is reported after the conclusion.